# Association of Rheumatoid Arthritis with Opioid Pain Medication Overuse among Persons Exposed to the 9/11 World Trade Center Disaster

**DOI:** 10.3390/ijerph20054166

**Published:** 2023-02-25

**Authors:** Ananya Sarker Dhanya, Janette Yung, James E. Cone, Jiehui Li

**Affiliations:** 1New York City Department of Health and Mental Hygiene, World Trade Center Health Registry, Long Island City, NY 11101, USA; 2Department of Epidemiology and Biostatistics, City University of New York (CUNY) Graduate School of Public Health & Health Policy, 55W 125th Street, New York, NY 10027, USA

**Keywords:** World Trade Center (WTC) disaster, rheumatoid arthritis, posttraumatic stress disorder (PTSD), opioids, pain medication

## Abstract

We examined the association of post-9/11 rheumatoid arthritis (RA) diagnosis with opioid pain medication overuse among enrollees in the World Trade Center Health Registry (WTCHR). Opioid overuse was defined as the self-reported intake of prescribed opioids at a higher dosage or more often than directed in the last 12 months on one of the two most recent WTCHR surveys (2015–2016, 2020–2021). Post-9/11 RA was ascertained through self-reports and subsequently validated following medical record release by the enrollees’ physicians or medical records review. We excluded those with self-reported RA that was not validated by their physicians and those who did not report being prescribed opioid pain medication in the last 12 months. Multivariable log-binomial regression was conducted to examine the relationship between post-9/11 RA diagnosis and opioid pain medication overuse, adjusting for sociodemographic characteristics and 9/11-related posttraumatic stress disorder (PTSD) symptoms. Of the 10,196 study enrollees, 46 had confirmed post-9/11 RA. The post-9/11 RA patients were mostly females (69.6% vs. 37.7%), less frequently non-Hispanic White (58.7% vs. 73.2%) individuals, and less often had attained a higher level of education (76.1% vs. 84.4%) compared to those without post-9/11 RA. Opioid pain medication overuse was significantly associated with a post-9/11 RA diagnosis (Adjusted Risk Ratio: 2.13, 95% CI: 1.44–3.17). More research is needed to better understand the use and management of prescribed opioids among WTC-exposed individuals with RA.

## 1. Introduction

The 11 September 2001 terrorist attack on the World Trade Center (WTC) in New York City (NYC) exposed hundreds of thousands of individuals to horrific events and potentially harmful environmental contaminants and toxic substances resulting from the collapsing towers and fires [1,2,3]. Systemic autoimmune diseases (SADs), including rheumatoid arthritis (RA), Sjogren’s syndrome, systemic lupus erythematosus, mixed connective tissue disease, polymyositis, and scleroderma, are an emerging health concern among persons who were exposed to the WTC disaster [4,5]. Of all the reported SADs, RA is the most common diagnosis [4,5], which has been linked to the persistent use of pain medication, including opioids [6]. However, it is unknown to what extent opioid pain medication is overused among WTC-exposed individuals who were diagnosed with RA.

RA is a heterogeneous SAD that mainly affects the diarthrodial joints involving cartilage at the articular surfaces, ligaments that form a joint capsule, and synovial fluid [7,8]. RA is the most common form of inflammatory arthritis that can eventually cause cartilage and bone damage. It has a substantial societal effect in terms of cost, disability, and lost productivity [7,8,9]. The onset of RA may be insidious or acute, with the symptoms usually starting in the small joints of the hands and feet and including joint swelling, tenderness to palpation, morning stiffness, and severe motion impairment in the involved joints [10]. In the United States (U.S.), the prevalence of RA was about 0.5% between 2004 and 2014 [11]. Most causes of RA are unknown and are suggested to be the interplay between genetic, autoantibody, and environmental risk factors [12]. The disorder is more common in women and older adults [13], with smoking being the main behavioral or environmental risk factor [12,13].

There are various types of medications used for RA treatment. The first-line treatment includes disease-modifying antirheumatic drugs (DMARDs), such as methotrexate [14]. Other treatments commonly used to alleviate RA symptoms include glucocorticoids and the sequential application of biological agents, such as tumor necrosis factor inhibitors [14]. Another treatment alternative or supplement to alleviate the symptoms of RA is pain remediation. It has been found that adequate pain therapy is essential for disease management for elderly patients with RA [15]. There are two main groups of pain medications: non-steroid anti-inflammatory drugs (NSAIDs) and opioids. While NSAIDs can result in potentially serious gastrointestinal, cardiovascular, and renal adverse effects, particularly among the elderly [15], opioids provide a treatment option for those whose pain is poorly controlled or who have adverse reactions to NSAIDs.

Recent cohort studies among the U.S. population have estimated that up to 40% of patients with RA are regular users of opioids [16] and that the concurrent usage of other treatments, including DMARDs, did not reduce opioid use in RA patients [6]. There are different pharmaceutical forms of opioids available to treat pain, including naturally derived (e.g., morphine), semisynthetic (e.g., oxycodone and hydrocodone), and fully synthetic (e.g., fentanyl) opioids. The semisynthetic forms are widely used to treat chronic pain such as RA, whereas morphine and fentanyl are typically prescribed to treat severe pain following surgery or advanced-stage cancer [16,17,18]. Though the proven efficacy of short-term opioid use for symptom alleviation has been reported [16], there has been a rising concern regarding overuse and dependence considering opioids’ potentially addictive properties [19]. Chronic opioid use doubled in the U.S. between 2002 and 2015 among RA patients, with significant predictors including pain level, antidepressant use, high inflammatory disease activity, and the level of disability [20]. In 2019, the Association of Rheumatology Professionals (ARP) reported that RA patients were at a 1.5 to 2 times higher risk of opioid overdose-related hospitalization compared with the general population [21].

Opioid overuse is an initial phase of opioid use disorder (OUD), which could lead to serious consequences such as breathing difficulty and eventually death [22,23]. According to the National Institute on Drug Abuse (NIDA), opioid overdose death rates increased by 56% between 2010 and 2015 in the U.S. [24], despite a 13% reduction in opioid prescriptions for the same time period [24]. These data indicate that a decreasing trend of opioid prescription has not effectively prevented the occurrence of opioid overdose and associated deaths among the U.S. population. The increasing opioid-related death toll is largely due to the unregulated flood of illicit fentanyl into the market, which is a potent opioid that is 50 to 100 times stronger than morphine [25].

Significant physical and psychological morbidities were experienced by individuals after exposure to the WTC disaster following the terrorist attacks on 11 September 2001 (hereafter, 9/11) [26]. Of all the morbidities, posttraumatic stress disorder (PTSD) is a commonly reported mental health condition among the 9/11-exposed populations. An association between PTSD and the onset of RA has been reported in both the veteran and civilian populations [27,28]. Additionally, previous studies reported a strong bidirectional relationship between PTSD and substance use disorder, including opioid overuse, suggesting that the two conditions can reinforce each other [29,30]. Changes in social behaviors such as using substances to relieve symptoms and cope with discomfort temporarily have been documented among those who suffer from PTSD [31,32]. Patients with PTSD may have increased pain perception [27], which necessitates opioid pain medication for symptom alleviation [28]. A study of a WTC-exposed population showed that opioid pain medication overuse was more prevalent among those who have a history of PTSD, compared with those who never had PTSD [32]. Therefore, it is important to account for this co-morbidity when assessing opioid overuse among RA patients who were exposed to the WTC disaster [32].

There are limited reports on opioid overuse among RA patients in the 9/11-exposed population. In the present study, we aimed to examine whether a post-9/11 RA diagnosis was associated with recent opioid pain medication overuse in a cohort of the WTC-exposed population, adjusting for sociodemographic characteristics and 9/11-related PTSD symptoms.

## 2. Materials and Methods

### 2.1. WTC Health Registry and Data Source

The World Trade Center Health Registry (WTCHR) was established in July 2002 to track and evaluate the health effects among the individuals exposed to the WTC disaster, including rescue and recovery workers (RRWs) and non-RRWs. It is by far the largest post-disaster registry in U.S. history [26]. A detailed description of the Registry’s recruitment methods has been published elsewhere [26,33]. In brief, the Registry enrolled over 71,000 participants in 2003–2004 [33]. Data on sociodemographic characteristics, WTC exposures, and medical (physical & mental) history were collected. All the enrollees provided verbal informed consent at enrollment (wave 1) so that their responses could be used for the data analyses [4,26]. Since enrollment, four subsequent follow-up surveys have been completed in 2006–2007 (wave 2), 2011–2012 (wave 3), 2015–2016 (wave 4), and 2020–2021 (wave 5) [33,34]. In addition to these full-wave surveys, the Registry also conducts in-depth surveys for various health conditions to enhance surveillance on newly emerging diseases. In this study, we used the data from wave 1, waves 3–5 and an in-depth survey of SADs [4].

### 2.2. Study Sample

Of the 71,424 total enrollees in the WTCHR, we excluded those who did not respond to the wave 4 and wave 5 surveys (*n*= 31,216), who were aged < 18 years at the time of 9/11/2001 (N = 1386), who had self-reported RA in wave 3 that was not confirmed by medical records or treating physicians (*n* = 2404), who had self-reported RA for the first time in either wave 4 or 5 without prior physician’s validation (*n* = 2029), and who had confirmed RA that was diagnosed prior to the WTCHR enrollment or with an unknown year of diagnosis (*n* = 8). Of the remaining 34,381 enrollees, we limited the analysis to those who reported having been prescribed any pain-relieving opioid medication (e.g., oxycodone or hydrocodone) in the preceding 12 consecutive months in either wave 4 or wave 5 (*n* = 10,777). Among these individuals, we further excluded those with a missing value in any of the covariates. The covariates with missing values include marital status (*n* = 114), education (N = 94), smoking history (*n* = 110), and missing items on any of the self-reported PTSD checklist assessments in both wave 4 and wave 5 (*n* = 321), all of which were from the non-RA comparison group, except for one enrollee with a missing marital status in the RA group. The final sample includes a total of 10,196 participants.

### 2.3. Outcome Pain Medication Overuse

The enrollees were considered as overusing opioid pain medication if they responded “Yes” to both the gate and follow-up questions regarding overuse behavior in their latest wave response (wave 5 or wave 4). The gate question asked, “During the last 12 months, have you ever—even once—taken the pain reliever that you were prescribed (Oxycodone (e.g., Percocet, Endocet, OxyContin) or Hydrocodone (e.g., Vicodin, Norco, Lortab))? Do not include the “over the counter” medications”. If they responded “Yes” to the query, a follow-up question was asked to inquire about whether they had ever taken that prescribed pain medication at a higher dosage or more often than directed in the last 12 months. The enrollees were considered not to have overused opioid pain medication if they answered “No” to either the gate or the follow-up question.

### 2.4. RA Ascertainment

The RA ascertainment was obtained from an in-depth survey of SADs among Registry adults who reported having RA or any other SADs since 2001 in wave 3 [4]. The details of the in-depth survey were described elsewhere [4]. In brief, all the wave 3 participants who responded positively to a question asking whether they were ever diagnosed with any SADs, including RA, were further studied to confirm their reported diagnosis. The survey also asked whether they were taking a medication consistent with treatment for SADs and sought their permission to validate their self-reported diagnoses by sending a physician survey to their rheumatologists or the physicians who could either provide the clinical information regarding their autoimmune disease diagnosis or send their medical records directly to the Registry for abstraction [4]. RA cases were confirmed if at least one of the following three criteria was met: (1) the American College of Rheumatology (ACR) score-based algorithm with a score of ≥6/10 based on the categories of joint involvement, serology, acute-phase reactants, and the duration of symptoms [35]; (2) received a prescription for medication consistent with an autoimmune disease; or (3) received a RA diagnosis by a board-certified rheumatologist [35]. Documentation showing a positive response to any of the three criteria was reviewed independently by two Registry research staff, including the Registry’s Medical Director (SM-A, JEC) [4]. Unclear diagnoses were reviewed independently by two board-certified rheumatologists, and upon any disagreement, they met and conferred until they agreed [4]. All the clinically confirmed RA cases diagnosed after enrollment were defined as having post-9/11 RA (hereafter, RA). The enrollees who never reported having a diagnosis of RA in any of the follow-up surveys were considered to be without RA.

### 2.5. Sociodemographic Characteristics

The sociodemographic characteristics included the age at 9/11 (18–39, 40–49, 50–79 years), sex, race/ethnicity (non-Hispanic White, non-Hispanic Black, Hispanic, Asian, and non-Hispanic all other races (includes Multiracial, American Indian, Alaskan Native, unknowns)), marital status (married/partnered vs. other (never married, widowed, or divorced/separated)), educational attainment (≤high school/General Educational Development (GED) graduate vs. >high school (some college, college graduate, graduate degree)), and smoking history (never, former, current). The data on age, sex, and race/ethnicity were collected at baseline, whereas marital status, education level, and smoking status were obtained from either wave 4 or wave 5 survey responses, consistent with the source from which the outcome was used.

### 2.6. 9/11-Related PTSD Symptoms

9/11-related PTSD symptoms (hereafter, PTSD) were assessed from enrollees’ latest response (wave 5 or wave 4) using the PTSD Checklist-Specific (PCL-S), a 20-item (Diagnostic and Statistical Manual for Mental Disorders (DSM-V) criteria) or 17-item (DSM-IV criteria) self-reported 9/11 event-specific questionnaire from the American Psychiatric Association (APA), respectively [32]. The original DSM-IV measure in wave 4 was adapted to the DSM-V (wave 5) scale [36]. A score of 33 or higher (out of a total of 80) on the questionnaire was considered as having PTSD symptoms [37]. The participants were categorized as either having or not having active PTSD. This method had been shown to have good validity and reliability, with an internal consistency of 96% and a test-retest reliability of 84% [36,38].

### 2.7. Statistical Analyses

We conducted descriptive analyses to explore the sociodemographic characteristics, PTSD, and opioid pain medication overuse among those with and without an RA diagnosis. We examined the bivariate relationship of post-9/11 RA diagnosis, 9/11-related PTSD symptoms, and all other covariates with opioid pain medication overuse using chi-square statistics. Next, we ran both unadjusted and adjusted log-binomial regression models to examine the association between a post-9/11 RA diagnosis and pain medication overuse. All the statistical tests were 2-sided, and a *p*-value less than 0.05 was considered statistically significant. All the analyses were conducted in Statistical analysis software (SAS) Enterprise Guide (version 7.15, SAS Institute Inc., Cary, NC, USA).

## 3. Results

Of the 10,196 enrollees included in the analysis, 46 (0.5%) had post-9/11 RA confirmed by clinical information (Table 1). Overall, a higher proportion of those with RA were of middle age (40–49) on 9/11 (47.8% vs. 35.4%), were female (69.6% vs. 37.7%), had PTSD (28.3% vs. 14.8%), and had opioid pain medication overuse (28.3% vs. 11.1%) compared to those without RA. In contrast, a higher proportion of those without RA were non-Hispanic White (73.2% vs. 58.7%) individuals, had greater than high school education (84.4% vs. 76.1%), and were married/living with a partner (70.0% vs. 65.2%) compared with those with RA.

The results of the bivariate analyses of RA, PTSD, and the covariates with recent opioid pain medication overuse are presented in Table 2. Of the 10,196 study enrollees, 1141 (11.2%) reported recent overuse of opioid pain medication. The proportions of those with RA or PTSD were greater among those who reported overusing opioid pain medication compared to those who did not (RA: 1.1% vs. 0.4%, *p* = 0.0002; PTSD: 31.8% vs. 12.7%, *p* < 0.0001, respectively).

Table 3 shows both the unadjusted and adjusted results of the log-binomial regression models for the association between post-9/11 RA diagnosis and self-reported opioid pain medication overuse. In the unadjusted model, post-9/11 RA diagnosis was associated with recent pain medication overuse (unadjusted risk ratio (URR): 2.54, 95% CI: 1.60–4.04). In the multivariable model, the adjusted risk ratio (ARR) for RA was slightly attenuated but remained significant after adjusting for PTSD and other confounders (ARR: 2.13, 95% CI: 1.44–3.17).

## 4. Discussion

In the present study, we observed a statistically significant association between RA and opioid pain medication overuse among WTCHR enrollees, after adjusting for PTSD and sociodemographic characteristics. Our finding is consistent with the finding from a previous population-based comparative study that reported that patients with RA have a higher rate of opioid use compared with patients without RA [6]. An increased prevalence of chronic opioid use among RA patients or persons with arthritis has been reported for the period of 2002–2015 [20,39]. Our study highlights the importance of the close monitoring of opioid medication use among RA patients.

Opioid pain medication overuse was more common among our enrollees with RA than those without RA (28.3% vs. 11.1%). A possible biological mechanism is that RA patients with a long-term opioid usage history have increased pain perception due to central sensitization of the pain modulation system in the central nervous system, which eventually results in dependency and an incremental increase in opioid use [40,41]. In this study, all the RA cases were diagnosed prior to wave 3 in 2011–2012, which could indicate that our RA patients had at least 4 years of the illness and were potentially more likely to use of pain medication as time elapsed since wave 3. In the U.S., the overall opioid overdose death rates have also increased by 56% from 2010 to 2015 [24,26]. Similar trends were also reported in New York State and New York City [42,43], where the majority of WTCHR enrollees reside. Given the serious consequences of overuse, such as disordered breathing [22,23], and the vulnerability of our WTC-exposed population due to a higher prevalence of exposure-related physical and mental health conditions than the general population [44], it is important to evaluate the risk of overuse when considering opioid as a pain management option among RA patients [45]. More research is necessary to evaluate the long-term pattern of opioid use among this population.

The association between RA and opioid pain medication overuse attenuated slightly and remained significant after adjusting for PTSD status, suggesting that our observed association between RA and opioid pain medication overuse was independent of PTSD, despite the fact that PTSD has been found to be a risk factor for opioid overuse [30,46]. Previous studies reported an association between PTSD and the onset of RA [27,28]. It has been suggested that PTSD heightens systemic inflammation by disrupting a major hormone regulatory system, the hypothalamic–pituitary–adrenal (HPA) axis [47,48], which can result in the development of RA or worsening of the disease progression [47,49,50]. The higher prevalence of PTSD in our population compared with the general population [32] may result in the increased pain in those with RA, which may subsequently lead to a higher rate of opioid medication use for pain management. Therefore, as part of a holistic approach, it is essential for physicians to screen for PTSD and the risk of opioid overuse during routine health exams for RA patients who were exposed to the WTC disaster or any other trauma [45].

The U.S. government has taken various initiatives to restrict supply and reduce demand to address the excessive use of opioids. The strategies that were found to be effective include regulating the approved product, restricting lawful access to approved drugs, providing education and prescribing guidelines at the provider, patient, and public level, and a prescription monitoring program (PMP) to track access to the various medical treatment options causing OUD [51,52]. Adherence to these strategies is essential to prevent and control the increase in pain medication overuse in vulnerable populations.

### Limitations

This study is not without limitations. First, we cannot discount attrition bias, as a major (63%) portion of the enrollees with RA did not participate in the follow-up in-depth survey or did not provide consent for the providers’ survey [4]. Moreover, previous Registry analyses showed that enrollees with intermittent survey responses were more likely to have had poorer health or experienced PTSD compared with those who completed all waves [53]. Compared to those who were not included in our analysis, our sample had a higher proportion of non-Hispanic White (72.8% vs. 54.7%) individuals and those with more than a high school education (84.0% vs. 71.5%). It is possible that those enrollees who were not in our study might have a different pain medication usage pattern than our study sample. Second, the small number of RA enrollees may lower the statistical power of our analysis. Furthermore, the non-RA status in our comparison group was based on a self-report without clinical validation, which may have led to the inclusion of the unreported or undiagnosed RA in this group and resulted in an underestimation of our findings. Third, we do not have data on compliance regarding prescribed pain medication use and the reasons for overuse, which may bias our observed findings. Moreover, PTSD was self-reported and not clinically confirmed, subjecting the analysis to recall bias. Lastly, though it is well known that a family history of RA has a strong influence on RA diagnosis as well as its severity [54], we were unable to control for this factor due to a lack of family history data. It is also possible that some other factors that were not available in this study may have contributed to opioid pain medication overuse in this population, such as a personal history of substance misuse, including alcohol, tobacco, and marijuana [23,55].

## 5. Conclusions

Our study demonstrated that RA was associated with an increased risk of opioid pain medication overuse among WTCHR enrollees. It is crucial for clinicians to closely monitor the use of prescribed opioids among RA patients with prior exposure to traumatic events, such as the WTC disaster. Additionally, more research and periodic evaluation are warranted to better understand the long-term use and management of prescribed opioids among WTC-exposed individuals with RA. Intervention(s) targeting opioid-related harm reduction, such as overuse education and PMP [51,52], to reduce the risk of prescribed opioid overuse, may be recommended.

## Figures and Tables

**Table 1 ijerph-20-04166-t001:** Sociodemographic characteristics, active PTSD symptoms, and pain medication overuse by post-9/11 rheumatoid arthritis (RA) among WTCHR enrollees ^1^ (N = 10,196).

	Enrollees with Post-9/11 RA	Enrollees without Post-9/11 RA
Characteristics	N (%)	N (%)
Total	46	10150
Age at 9/11, years		
18–39	14 (30.4)	3941 (38.8)
40–49	22 (47.8)	3595 (35.4)
50–79	10 (21.7)	2614 (25.8)
Gender		
Male	14 (30.4)	6324 (62.3)
Female	32 (69.6)	3826 (37.7)
Race/Ethnicity ^2^		
Non-Hispanic White	27 (58.7)	7433 (73.2)
Non-Hispanic Black	9 (19.6)	965 (9.5)
Hispanic Asian	10 (21.7) 0 (0.0)	1087 (10.7) 343 (3.4)
Non-Hispanic, all other races	0 (0.0)	322 (3.2)
Marital status ^3^		
Married or living with a partner	30 (65.2)	7110 (70.0)
All other	16 (34.8)	3040 (30.0)
Educational attainment		
High school graduate/GED	11 (23.9)	1581 (15.6)
>High school graduate	35 (76.1)	8569 (84.4)
Smoking history ^4^		
Current smoker	4 (8.7)	914 (9.0)
Former smoker	17 (37.0)	3439 (33.9)
Never smoker	25 (54.3)	5797 (57.1)
Active PTSD symptoms ^5^		
Yes	13 (28.3)	1504 (14.8)
No	33 (71.7)	8646 (85.2)
Opioid pain medication overuse		
Yes	13(28.3)	1128 (11.1)
No	33 (71.7)	9022 (88.9)

WTCHR = World Trade Center Health Registry; PTSD = posttraumatic stress disorder; GED = General Educational Development. ^1^ Sample excludes enrollees who were younger than age 18 on 9/11, did not respond to either wave 4 or wave 5, had a self-reported RA diagnosis, were diagnosed with RA prior to 9/11, had a diagnosis of RA prior to enrollment; sample only included enrollees who were prescribed opioid pain medication within the preceding 12 consecutive months of either wave 4 or wave 5. ^2^ “Non-Hispanic, all other races” includes Multiracial, American Indian, Alaskan Native, unknowns. ^3^ “All other” includes never married, divorced, separated, or widowed. ^4^ Never smoker = Had not smoked ≥ 100 cigarettes in lifetime, Former smoker = Had smoked ≥ 100 cigarettes in lifetime previously and no longer smoked, Current smoker = Had smoked ≥ 100 cigarettes in a lifetime and a current smoker at the time of the survey. ^5^ “Active PTSD symptoms” was assessed from the enrollees’ latest wave response (wave 5 or wave 4), categorized based on the DSM-V criteria of Probable PTSD symptoms (Probable PTSD if 33 or more out of 80).

**Table 2 ijerph-20-04166-t002:** Sociodemographic characteristics, rheumatoid arthritis (RA), and 9/11-related active PTSD symptoms by pain medication overuse among WTCHR enrollees ^1^ (N = 10,196).

	Total	Opioid Pain Medication Overuse	No Opioid Pain Medication Overuse	*p*-Value
Characteristics	N (%)	N (%)	N (%)	
Total	10196	1141 (11.2)	9055 (88.8)	
Age on 9/11, years				
18–39	3955 (38.8)	517 (45.3)	3438 (38.0)	<0.0001 **
40–49	3617 (35.5)	432 (37.9)	3185 (35.2)	
50–79	2624 (25.7)	192 (16.8)	2432 (26.9)	
Gender				
Male	6338 (62.1)	740 (64.9)	5598 (61.8)	0.0465 *
Female	3858 (37.9)	401 (35.1)	3457 (38.2)	
Race/Ethnicity				
Non-Hispanic White	7460 (73.2)	770 (67.5)	6690 (73.9)	<0.0001 **
Non-Hispanic Black	974 (9.5)	108 (9.5)	866 (9.6)	
Hispanic	1097 (10.8)	163 (14.3)	934 (10.3)	
Asian	343 (3.4)	59 (5.2)	284 (3.1)	
Non-Hispanic, all other races	322 (3.2)	41 (3.6)	281 (3.1)	
Marital status				
Married or living with a partner	7140 (70.0)	754 (66.1)	6386 (70.5)	0.0020 *
All other	3056 (30.0)	387 (33.9)	2669 (29.5)	
Educational attainment				
High school graduate/GED	1592 (15.6)	239 (20.9)	1353 (14.9)	<0.0001 **
More than high school graduate	8604 (84.4)	902 (79.1)	7702 (85.1)	
Smoking history				
Current smoker Former smoker Never smoker	918 (9.0) 3456 (33.9) 5822 (57.1)	199 (17.4) 367 (32.2) 575 (50.4)	719 (7.9) 3089 (34.1) 5247 (58.0)	<0.0001 **
Active PTSD symptoms				
Yes	1517 (14.9)	363 (31.8)	1154 (12.7)	<0.0001 **
No	8679 (85.1)	778 (68.2)	7901 (87.3)	
Post-9/11 Rheumatoid Arthritis Yes	46 (0.5)	13 (1.1)	33 (0.4)	0.0002 **
No	10,150 (99.5)	1128 (98.9)	9022 (99.6)	

WTCHR = World Trade Center Health Registry; PTSD = posttraumatic stress disorder; GED = General Educational Development. *p* value using Chi-squared test— * ≤ 0.05, ** ≤ 0.001. ^1^ Sample excludes enrollees younger than age 18 on 9/11, those who did not respond to either wave 4 or wave 5, those with an unvalidated self-reported RA diagnosis, those diagnosed with RA prior to 9/11, and those with a diagnosis of RA prior to enrollment; sample only included enrollees who were prescribed opioid pain medication within the preceding 12 consecutive months of either wave 4 or wave 5.

**Table 3 ijerph-20-04166-t003:** Unadjusted and adjusted relative risk (RR) and 95% confidence interval (CI) for the association of risk factors with pain medication overuse among WTCHR enrollees ^1^ (N = 10,196).

Characteristics	URR	95% CI	ARR	95% CI
Rheumatoid Arthritis				
Post-9/11 RA No RA	2.54 referent	1.60, 4.04	2.13 referent	1.44, 3.17
Active PTSD symptoms				
Yes No	2.67 referent	2.39, 2.99	2.27 referent	2.03, 2.56
Age on 9/11, years				
18–39	1.79	1.53, 2.09	1.61	1.37, 1.89
40–49	1.63	1.39, 1.92	1.48	1.26, 1.74
50–79	referent		referent	
Race/Ethnicity				
Non-Hispanic Black	1.07	0.89, 1.30	1.05	0.87, 1.27
Hispanic Asian	1.44 1.67	1.23, 1.68 1.31, 2.12	1.13 1.63	0.97, 1.32 1.29, 2.04
Non-Hispanic, all other races Non-Hispanic White	1.23 referent	0.92, 1.65	1.11 referent	0.83, 1.47
Gender				
Male	1.12	1.00, 1.26	1.18	1.05, 1.33
Female	referent		referent	
Marital status				
All other Married or living with a partner	1.20 referent	1.07, 1.35	1.13 referent	1.00, 1.27
Educational attainment				
High school graduate/GED	1.43	1.25, 1.63	1.21	1.07, 1.38
More than high school graduate	referent		referent	
Smoking history				
Current smoker	2.19	1.90, 2.54	1.80	1.56, 2.08
Former smoker	1.08	0.95, 1.22	1.18	1.05, 1.34
Never smoker	referent		referent	

WTCHR = World Trade Center Health Registry; URR = unadjusted risk ratio; ARR = adjusted risk ratio; PTSD = posttraumatic stress disorder; GED = General Educational Development. ^1^ Sample excludes enrollees younger than age 18 on 9/11, those who did not respond to either wave 4 or wave 5, those with an unvalidated self-reported RA diagnosis, those diagnosed with RA prior to 9/11, and those with a diagnosis of RA prior to enrollment; sample only included enrollees who were prescribed opioid pain medication within the preceding 12 consecutive months of either wave 4 or wave 5.

## Data Availability

The data presented in this study are available on request from the corresponding author.

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
