# Peer review of "Association of Rheumatoid Arthritis with Opioid Pain Medication Overuse among Persons Exposed to the 9/11 World Trade Center Disaster"

_ijerph, 2023, doi:10.3390/ijerph20054166_

Round 1

Reviewer 1 Report

Comments

1.The article is  a survey report. Its mere collection of information from various resource.

2.Validation of such report must have been  included in the study.

3.What is 9/11-exposed populations. It should be written “9/11 years” instead of “9/11”.

4. How author arrived only 9/11 years population to be most effective. There is no confirmation and validation report.

5. What are the necessary steps attempted by the country government to reduce the affected population by consumption of Opioid Pain Medication. Reports must have been included

6. It is necessary to generate report for future population to reduce/nullify the consumption of Opioid Pain Medication.

7. The concluding message in  this report is not very clear.

Reviewer 2 Report

Dear Authors,

I read with great interest your paper entitled "Association of Rheumatoid Arthritis with Opioid Pain Medication Overuse among 9/11-exposed Population". I have a few suggestions that I feel can improve the text:
- in the introduction, I would suggest to describe the relationship between e.g. PTSD (or more broadly - psychopathology) and substance abuse. PTSD is one of the controlled variables in the study and, in my opinion, requires a broader description.
- in the text, we can read about two waves of research - was that a significant factor? How did it change between the two measurements?

Thank you very much for the opportunity to read this manuscript.

Reviewer 3 Report

In their article entitled “Association of Rheumatoid Arthritis with Opioid Pain Medication Overuse among 9/11-exposed Population,” Dhanya et al. examine the relationship between rheumatoid arthritis and opioid pain medication overuse in a cohort of individuals who experienced the traumatic events of the September 11, 2001, World Trade Center disaster. Using the World Trade Center Health Registry data set, Dhanya et al. conducted multivariable log-binomial regression analyses while controlling for sociodemographic characteristics and PTSD symptoms. The authors found that opioid overuse was significantly associated with post-9/11 rheumatoid arthritis diagnosis independent of PTSD.

Apart from minor grammatical and formatting errors, this paper was concise and easy to interpret. There are a few ways in which the overall introduction and discussion of the findings could be improved to better frame and dig deeper into the results, however.

In the introduction, the transition to framing RA in the context of post-9/11 is unclear. Is the association between opioid overuse and RA specifically within the WTCHR important from the perspective of PTSD, or is it from the general association of the two? I wonder about this again in the discussion when the association exists independently of PTSD. In other words, does the novelty of this analysis of the relationship between RA and opioid overuse stem from the additional variable of such a traumatic event? If yes, I think the introduction could be strengthened by focusing on the nice sentence spanning lines 93-94.

From the surveys given to the enrollees, were there any questions pertaining to a family history of RA? Family histories of substance use are mentioned in the limitations section, but what about RA? Any information regarding family history of RA or SADs would be helpful information to include if possible.

In the legend for Table 2 (lines 208-209), is the “not current smoker, Current smoker = …lifetime; current smoker.” correct?  The categories mentioned in the text were “never, former, current,” so I am confused by the meaning of this line in the figure legend.

In the discussion, could the authors expound upon the existing literature with which the findings are consistent (lines 246-247)? As this is the discussion section, framing the results in the context of the existing literature would improve the section overall.

From line 258 in the discussion, is it particularly important to evaluate risk in post-9/11 individuals only, or should physicians consider any past traumatic experiences of individuals? Further, if RA and opioid overuse were related independently of PTSD, then should PTSD screenings be essential (Lines 268-270)?

While the conclusion does a nice job summarizing the findings of the study and putting the results in a larger context, the way in which the discussion is written does not seem to arrive at the same conclusions. The discussion section could be improved by applying the findings on a larger scale, much like what the authors have in the conclusion. In reading the sections, it seemed that the discussion section made one conclusion regarding specific treatment of WTCHR enrollees while the conclusion section itself introduced the arguably more applicable ideas stated in lines 297-299 for RA and opioid overuse prevention as a whole.

Round 2

Reviewer 1 Report

Author provided considerable improved manuscript  in-spite of several limitations in  this study. However few minor clarification will make the manuscript stronger.

1.Author validated the survey by consulting doctors prescription regarding the response of patient. It is important to know whether patient followed consumption of drugs strictly as per the doctors prescription or stopped or taken overdose due to unbearable pain without doctors consultation etc. The response to this comment will be make the study more stronger.

2.Geographical map showing the more affected area of US due to opioid pain medication overuse among WTCHR enrolees, Opioid overdose death rates, unregulated flood of illicit fentanyl etc and also  significant  improvement  area after prevention of  illicit fentanyl/opioid supply,  mentioning the year and other information   may be provided for better visibility of the conducted study.

3.In-spite of  limitation, could author collected some consents. Please provide information that how many consents were collected out of how many enrolees.   

4.Full form of abbreviation SADs, SAS must be provided.
